



# The response of ionospheric currents to different types of magnetospheric fast flow bursts using THEMIS observations

Homayon Aryan[1], Jacob Bortnik[1], Jinxing Li[1], James Michael Weygand[2], Xiangning Chu[3], and Vassilis Angelopoulos[2]

[1]University of California Los Angeles, Atmospheric and Oceanic Sciences, Math Sciences Building, Los Angeles, CA 90095-1565, United States.
[2]Department of Earth, Planetary and Space Sciences, University of California Los Angeles, California 90095, USA.
[3]Laboratory for Atmospheric and Space Physics, University of Colorado, Boulder, Colorado 80303, USA.

**Correspondence:** Homayon Aryan (aryan.homayon@gmail.com)

**Abstract.**

The magnetotail earthward fast flow bursts can transport most of the magnetic flux and energy into the inner magnetosphere. These fast flow bursts are generally an order of magnitude higher than the typical convection speeds, that are azimuthally local-

ized (1-3RE) and are flanked by plasma vortices which map to ionospheric plasma vortices of the same sense of rotation. This study uses multipoint analysis of conjugate magnetospheric and ionospheric observations to investigate the magnetospheric and ionospheric responses to the fast flow bursts that are associated with both substorms and pseudobreakups. We study in detail what properties control the differences in the magnetosphere-ionosphere responses between substorm and pseudobreakup conditions, and how such differences lead to the different ionospheric responses. The fast flow bursts and pseudobreakup events

were observed by the Time History of Events and Macroscale Interaction during Substorms (THEMIS), when the satellites were at least 6RE from the Earth in radial distance, and a magnetic local time (MLT) region of $\pm 5$ hours from local midnight. The results show that the magnetosphere and ionosphere response to substorm fast flow bursts are much stronger and more structured compared to pseudobreakups, which is more likely to be localized, transient, and weak in the magnetosphere. The magnetic flux in the tail is much stronger for strong substorms and much weaker for pseudobreakup events. The $B_{lobe}$ decreases

significantly for substorm fast flow bursts compared to pseudobreakup events. The curvature force density for pseudobreakups are much smaller than substorm fast flow events, indicating that the pseudobreakups may not be able to penetrate deep into the inner magnetosphere. This association can help us study the properties and activity of the magnetospheric earthward flow vortices from ground data.

## 1 Introduction

Magnetic reconnection in the Earth's magnetosphere converts open magnetic flux in the lobes into closed magnetic flux in the plasmasheet. A magnetospheric substorm is an important energy unloading processes in the magnetosphere. This process converts lobe magnetic energy into the thermal and kinetic energy of fast flow bursts, that are also known as bursty bulk





flows (BBFs), in the central plasmasheet Hones Jr. et al. (1970); McPherron (1970); McPherron et al. (1973); Baker (1996); Angelopoulos et al. (1992); Angelopoulos et al. (2008). These processes may repeat many times in the course of a moderate-
to-strong substorm Sergeev et al. (2014). These bursty bulk flows are an important component of plasmasheet dynamics during many different geomagnetic activity conditions. They are a common feature of radial transport throughout the plasmasheet and typically are associated with magnetic field dipolarizations Nakamura et al. (2002) and plasmasheet heating Runov et al. (2015). They are observed on short timescales of around minutes, and small scale sizes of a few Earth radius (RE) in the X and Y directions Gabrielse et al. (2019); Liu et al. (2013b). They exhibit large earthward velocities that are usually an order
of magnitude higher than the typical convection speeds and transport magnetic flux and energy into the inner magnetosphere that often decelerate and stop at around 8-10RE Angelopoulos et al. (1992); McPherron et al. (2011); Hsu and McPherron (2012); Runov et al. (2014); Sergeev et al. (2014); Liu et al. (2014). The rebound of earthward fast flow bursts can also cause tailward fast flow Nakamura et al. (2009); Birn et al. (2011). Even though substorms are closely associated with fast flow bursts generated by magnetic reconnection in the tail Liu et al. (2013a); Liu et al. (2015), not all fast flow bursts are
necessarily associated with a global response (e.g., McPherron2011, McPherronChu2018, Chu2015a) and they can also form spontaneously Sitnov et al. (2013). It was shown that the strength of a substorm is related to the magnetic flux accumulated in the inner magnetosphere Chu et al. (2021). Furthermore, the MPB index is used to distinguish the difference between global substorms and pseudobreakups Chu et al. (2015a). It is insensitive to the localized fine structure of the electrojet and can well capture the global substorm current wedge. Using a list of global substorms and pseudo-breakup, it was found that
substorm-onset-related fast flows are associated with stronger dipolarizations in Bz, and larger magnetic flux transport rates than non-substorm fast flows Li et al. (2021). Many fast flow bursts are associated with localized, transient, and weak response in the magnetosphere and ionosphere Li et al. (2021) and may not involve auroral brightening Nishimura et al. (2011) or plasmasheet injections (narrow high-speed flow bursts that were initially studied in detail as a substorm phenomenon Baker et al. (1982); Baker (1996); Lopez et al. (1990); McIlwain (1972) at geosynchronous orbit (GEO) due to the availability of
many satellite observations in this region) Akasofu (1964); Birn et al. (1997). However, substorm fast flow bursts are more likely to penetrate closer to the Earth, and are typically accompanied by a larger magnetic field increase and magnetic field energy input than non-substorm fast flow bursts Li et al. (2021).

    The magnetospheric substorm disturbances associated with the formation of the Substorm Current Wedge (SCW) Birn and Hesse (2014); Kepko et al. (2014); Kepko et al. (2015); McPherron (1979) that electrically couples the near-Earth plasmasheet
with the ionosphere through at least one pair of downward and upward Field-Aligned Currents (FACs) Liu et al. (2013a); Sun et al. (2013). The pressure gradient current and inertial current, are thought to be the sources of the SCW Yao et al. (2012); Birn and Hesse (2013), both of which are perpendicular currents and are believed to be diverted to form the field-aligned portion of the SCW Keiling et al. (2009). The field-aligned currents within the ionosphere are connected to one another via Pedersen currents. The westward electrojet and eastward electrojet are Hall currents that are believed to be anti-parallel to
the ionospheric convection. The auroral brightening, associated with substorm onset, is the deposition of electrons into the ionosphere Akasofu (1964); McPherron et al. (1973); Lyons et al. (2012), typically associated with the upward field-aligned current, as the magnetic field lines become more dipolarised and the SCW intensifies Chu et al. (2015).





The magnetosphere-ionosphere responses between substorm and non-substorm (pseudobreakups) conditions have been studied in the past (e.g., Ohtani et al. (1993); Koskinen et al. (1993); Nakamura et al. (1994); Baumjohann et al. (1989); Baumjohann

et al. (2010)). They have concluded that substorms and pseudobreakups have common responses (e.g., fast flows, dipolarizations, injections, electrojet and current wedge) without phenomenological differences. The differences between substorms and pseudobreakups are thought to be the strength, scale size and duration of activity; substorms have stronger and global activity but non-substorm conditions have weaker and localised activity. It has been shown that the substorm-time ionospheric currents have clockwise and counter-clockwise vortices Keiling et al. (2009) that are connected to plasma flow vortices in the

magnetosphere Akasofu (1976); Borovsky and Bonnell (2001). However, there are limited direct observational evidence of this connection largely due to the difficulty of finding conjunctions. Keiling2009 performed a multipoint analysis of conjugate magnetospheric and ionospheric flow vortices for a single substorm related fast flow bursts to show that the equivalent ionospheric currents (EIC) vortices were directly driven by the vortices observed in the magnetosphere. This study uses multipoint analysis of conjugate magnetospheric and ionospheric observations to investigate the magnetospheric and ionospheric responses to fast

flow bursts that are associated to both substorms and pseudobreakups.

In this study, we look into what properties control the differences in the magnetosphere-ionosphere responses between substorm and pseudobreakup conditions, and how such differences lead to the different ionospheric responses. We analyze the Time History of Events and Macroscale Interaction during Substorms (THEMIS) observations and select 3 pairs of fast flow and pseudobreakup events that were observed by all 3 inner THEMIS satellites (THEMIS A, D, and E) on the night side.

**2 Data**

The THEMIS mission Angelopoulos (2008) was launched in February 2007 and consists of five identically equipped satellites (A, B, C, D, and E). The main goal of this mission is to carry out multipoint investigations of substorm phenomena in the tail of the terrestrial magnetosphere Sibeck and Angelopoulos (2008). The fluxgate magnetometer (MAG) measures the background magnetic field Auster et al. (2008). The electric field instrument (EFI) Bonnell et al. (2008) measures the wave electric field.

The electrostatic analyzer (ESA) measures the thermal (5 eV - 25 keV) ions and electrons (McFadden et al., 2008). The solid state telescope (SST) measures the hot (25 keV to >1 MeV) ions and electrons Angelopoulos et al. (2008). In this study, 11 years (2008-2019) of observations from the 3 inner probes (A, D and E) are analyzed to identify fast flow bursts that were observed when the 3 satellites were closely separated on the night side, located at least 6RE away from the Earth in radial distance, and within a magnetic local time (MLT) region of ±5 hours from local midnight.

The all sky imager (ASI) data on the ground is analyzed to complement the response of ionosphere to fast flow bursts. The ground data provides contextual information on the processes observed in space by providing a two-dimensional view of the injection's formation and propagation, as well as its connection to the substorm evolution. A series of ground magnetometer arrays are used to generate the equivalent ionospheric currents (EICs) and current amplitudes at 10s resolution using the spherical elementary current systems (SECS) technique Amm and Viljanen (1999); Weygand et al. (2011, 2012); Weygand and

Wing (2016). They consist of a curl-free system whose divergences represent the FACs. It also consists of a divergence-free



elementary system that is contained entirely within the ionosphere. This study analyzes the EICs during conjunctions with the THEMIS satellites for the selected fast flow cases.

The electron flux data at geostationary orbit are measured by the Magnetospheric Electron Detector (MAGED) Rowland and Weigel (2012) onboard the Geostationary Operational Environmental Satellite (GOES) 13 and 15. The MAGED operates on five energy channels; 40 keV, 75 keV, 150 keV, 275 keV, and 475 keV and has nine telescopes pointing in different directions. This study investigates how electron flux data responds to different fast flow bursts.

The substorms are identified using the SuperMAG Auroral Electrojet Indices (SMU and SML) Gjerloev (2009); Newell and Gjerloev (2011) and the midlatitude positive bay (MPB) index McPherron and Chu (2017). The MPB index is calculated as the moving variance of changes in the H and D components ($\Delta H^2 + \Delta D^2$) obtained generally from 20 to 53 stations at midlatitudes ($20° - 52°$ in magnetic latitude) from the International Real-time Magnetic Observatory 127 Network Chu et al. (2015b). The MPB index is insensitive to the localized fine structure of the electrojet and can well capture the global substorm current wedge Chu et al. (2015b).

## 3 Selection Criteria

In this study, we analyzed the THEMIS observations to identify fast flow burst events that were observed by all 3 inner THEMIS satellites (THEMIS A, D, and E) during close separation on the night side, located at least 6RE away from the Earth in radial distance, and within a magnetic local time (MLT) region of ±5 hours from local midnight. The MPB index and the Auroral Electrojet Indices were used to distinguish between pseudobreakups (quiet conditions) and substorm (active conditions) fast flow bursts, where a MPB substorm is defined as the MPB index larger than 25nT$^2$. This study specifically searched for events that were observed at the end of 2015 when the THEMIS configuration was identical to that considered by Sergeev2012 was recreated by the THEMIS mission operations team. This unique configuration lasted around 3 months (October, November, and December) and allowed us to derive the time varying parameters such as the current density, lobe magnetic field, and plasma pressure. Three pairs of fast flow bursts, where one event represented pseudobreakups and another event represented substorm fast flow bursts. Also the fast flow burst pair were selected to occur within a few hours of each other on the same orbit, so that the background conditions were as similar as possible.

In addition, the fast flow bursts were selected based on at least one sample of the perpendicular velocity projected to the XY plane exceeding 150km/s. The start/end time of the fast flow bursts were defined by the first/last time when the earthward velocity component exceeded 120km/s. The observations while the THEMIS satellites were in the eclipse were excluded. Fast flow bursts that occurred within 60 seconds of each other were merged into one flow Li et al. (2021).

we need seperation along z to calculate the current



## 4 Observations

The THEMIS spacecraft in situ measurements for 6 selected fast flow burst cases are shown in Figure 1 to Figure 3. The pseudobreakups fast flow bursts are highlighted in green (Cases 1, 3, and 5) while substorm related fast flow bursts are highlighted in yellow (Cases 2, 4 and 6). Figure 1 shows the pair of fast flow bursts, Cases 1 and 2, that were observed on $25^{th}$ December 2015. The pseudobreakups fast flow burst, Case 1, was observed around 05:35 UT by all 3 THEMIS spacecraft. The MPB index was around $5nT^2$ and all 3 spacecraft observed magnetic field fluctuations. The Bx magnetic field component decreased by $\sim$20nT while the Bz magnetic field component increased by $\sim$20nT. THEMIS D observed the maximum ion perpendicular velocity, around 725km/s in the x direction. Approximately 2.5 hours later, at around 8:17 UT, all 3 THEMIS spacecraft observed another burst of fast flows, Case 2. The MPB index increased to just over $200nT^2$ and all 3 THEMIS spacecraft observed significant fluctuation in magnetic field and ion perpendicular velocity that were consistent across all 3 spacecraft. The Bx and Bz magnetic field components increased while THEMIS D observed the maximum ion perpendicular velocity of around 310km/s in the x direction.

Figure 2 shows the pair of fast flow bursts, Cases 3 and 4, that were observed on $20^{th}$ December 2015. The pseudobreakups fast flow burst, Case 3, was observed around 02:40 UT by all 3 THEMIS spacecraft when the MPB index was around $20nT^2$. All 3 spacecraft observed fluctuations in magnetic field and ion perpendicular velocity. The peak ion perpendicular velocity of around 290km/s was observed by THEMIS A while all 3 spacecraft observed an increase in the Bz magnetic field component. On the other hand, the substorm fast flow burst, Case 4, was observed around 2 hours later. The MPB index increased to more than $2500nT^2$, indicating a large substorm. All 3 THEMIS spacecraft observed significant variation in magnetic field and ion perpendicular velocity. The peak ion perpendicular velocity, around 530km/s, was observed by THEMIS E. In addition, Figure 3 shows the pair of fast flow bursts, Cases 5 and 6, that were observed on $10^{th}$ December 2015. The pseudobreakups fast flow burst, Case 5, was observed around 06:50 UT when the MPB index was around $10nT^2$. All 3 THEMIS spacecraft observed fluctuations in magnetic field and ion perpendicular velocity. In this case THEMIS E observed the largest peak ion velocity of around 600km/s. The substorm fast flow burst was observed less than an our later, around 07:40 UT, as the MPB index increased to around $570nT^2$. Again the fluctuations in the magnetic field and ion perpendicular velocity was consistently observed by all 3 THEMIS spacecraft. The Bx magnetic field components decreased while THEMIS D observed the maximum ion perpendicular velocity of around 500km/s in the x direction.

The location of the 3 THEMIS spacecraft (THEMIS A, D and E) in the GSM coordinate system for the selected fast flow bursts are shown in Figure 4. The yellow and green dots indicate where each fast flow was observed along the orbit of each THEMIS spacecraft. During the observation of these six fast flow burst events the configuration of the 3 THEMIS spacecraft were very favourable for determining the magnetic field gradients and plasma parameters in the magnetotail, because the y coordinates of the 3 satellites were almost the same, and it could be assumed that all differences between the magnetic field measured at the 3 satellites are caused by satellites separation in the (x, z) plane. In the tail science phase (September to December 2015 ) the apogee of the 3 THEMIS spacecraft were approximately 12 Re. The probes were separated by 1000 km to a few Earth radii at apogee. In addition, during the observation of the selected fast flow bursts, at least, one of the two GOES





13 and 15 satellites were ideally located on the night side to observe injections that may have been associated to the earthward

fast flow bursts. The next section discusses the derived equivalent ionospheric currents and current amplitudes, the ASI data on the ground, and the electron flux data from the MAGED measurements onboard the GOES 13 and 15 satellites associated with each fast flow burst.

The substorm ionospheric currents are typically accompanied with a clockwise and a counter-clockwise vortex associated with corresponding vortices in the magnetosphere. Multipoint analysis of conjugate magnetospheric and ionospheric flow

vortices for a single substorm related fast flow burst was performed by Keiling2009 to show that the EIC vortices were directly driven by the flow vortices in the magnetosphere. In this study, we investigate the magnetospheric and ionospheric response to fast flow bursts during both substorm and non-subtorm times. We analyzed in detail the six fast flow burst cases. Figure 5 shows the derived EICs and their current amplitudes plotted over ASI mosaics for fast flow burst Case 4 that was observed on 20$^{th}$ December 2015. This fast flow burst corresponds to Case 4 which is a substorm-time flow and the THEMIS satellites

began observing the flow around 04:50 UT on 20$^{th}$ December 2015 (Figure 1). Before the initiation of the flow burst, $\sim$ 04:45 UT (Figure 5a and 5b), there were very weak, large-scale clockwise flow vortices that overlapped with the foot print of the THEMIS satellites. The center of the vortex is located at about 55° Glat and 257° WGLong. Note that clockwise and counter-clockwise rotations correspond to downward and upward FAC, respectively. At this time there were only downward region 2 and upward region 1 currents. A two thin equatorward drifting east-west auroral arcs, moving south in ASIs WHIT (61° 225°),

FSIM (62° 239°), FSMI (60° 248°), and ATHA (55° 247°) were also observed starting at about 0445 UT. However, during the fast flow burst event, $\sim$ 04:50 UT (Figure 5c and 5d), relatively stronger counter-clockwise current vortices develops at 52° GLat and 265° WGlong and the downward region 1 current system and upward Harang current intensifies just north and south respectively of the THEMIS satellite foot points. An intensified westward electrojet and the poleward arc formed/brightened in ASIs FSIM and FSMI at about 0453 UT. The strong current vortices in the equivalent ionospheric currents were continuously

observed and intensified to the end of the fast flow burst, $\sim$ 04:58 UT (Figure 5e and 5f). A streamer in ASIs FSMI and ATHA is also observed starting at about 0453 UT and ending by about 0502 UT. The equivalent currents closest to the streamer point poleward from about 0453 UT to 0455 UT, then rotates and point toward the SW from 0457 UT to 0502 UT. The aurora brightened at the ATHA ASI from about 0451 UT and moved poleward into the field of view of the RANK ASI starting at 0456 UT. These observations are consistent with past observations Keiling et al. (2009); Li et al. (2021), indicating that the

ionospheric currents are associated with plasma flow vortices in the magnetosphere for fast flow bursts that are associated with substorms.

The same analysis was performed on pseudobreakups. Figure 6 shows the derived equivalent ionospheric currents and their current amplitudes plotted over a sequence of ASI mosaics for fast flow burst Case 1. This fast flow burst occurred during relatively quiet geomagnetic activity conditions and the inner THEMIS satellites began observing the flow around 05:35 UT

on 25$^{th}$ December 2015 (Figure 1). Before the initiation of this fast flow burst, $\sim$ 05:30 UT (Figure 6a and 6b), the EICs were very weak and the ASIs did not observe significant activity. However, during the fast flow burst, $\sim$ 05:38 UT (Figure 6c and 6d), larger EICs were observed near the satellite foot points. This enhancement in the EICs begins at about 0537 UT. To west of the spacecraft foot points the EICs point southward and to the east of the footpoints the EICs point poleward indicating the





foot points are within the Harang current system. The EICs continued to strengthen to the end of the fast flow burst, $\sim 05:50$

UT (Figure 6e and 6f). During the fast flow burst the spacecraft foot points were located in the upward (red) Harang current and between a downward region 1 current system and a downward region 2 current system. Starting at 0549 UT an auroral streamer appears between the downward and upward currents, consistent with the magnetospheric fast flow burst Kauristie et al. (2000); Nakamura et al. (2001, 2004). Figure 6e shows a streamer was present in both the GILL (56° 265°) and RANK (63° 268°) ASIs between 05:49 - 05:51 UT, but the poleward pointing EICS suggest that the streamer is present until about 0557 UT. The

EICs just to the east of the streamer pointed poleward, which is a good indicator of ionospheric flow from the North to the South and consistent with a north south streamer. The current density near the foot point of the THEMIS spacecraft just prior to the fast flow burst at 0535 UT had current density of about 0.1 $\mu A/m^2$ and increased to a peak value of 1 $\mu A/m^2$ near the end of the flow burst. This shows that although the ionosphere current response to the pseudobreakups is not as strong as it is for substorm fast flow bursts, nevertheless it is still connected to plasma flow vortices in the magnetosphere.

The earthward fast flow bursts could also play an important role in actively accelerating particles or directly injecting energetic particles into the inner magnetosphere. At substorm onset, the particle flux increases and often lasts tens of minutes to over an hour. In this study, we analyze the electron flux data from the MAGED observations onboard the GOES 13 and 15 satellites. These two satellites were ideally positioned on the night side to observe such injections for the selected fast flow burst cases.

The magnetosphere response to pseudobreakups, Case 1, at around 05:35 UT on 25[th] December 2015 is presented in Figures 7. The magnetic field measured by THEMIS A, D and E respectively is shown in panels a-c, while the perpendicular ion velocity measured by THEMIS A, D and E respectively is shown in panels d-f, and the magnetic field from GOES-13 and GOES-15 is shown in panels i and j respectively. The lower two panels (k and l) show the electron flux measured by GOES-13 and GOES-15 respectively. The grey vertical lines mark the peak perpendicular velocity associated with each fast flow burst

as observed by each THEMIS spacecraft. It is clear that both GOES-13 and GOES-15 satellites did not observe significant flux increase for any of the five energy channels (40 keV, 75 keV, 150 keV, 275 keV, and 475 keV). Also both GOES-13 and GOES-15 did not observe any significant variation in the magnetic field. For this particular case, both GOES-13 and GOES-15 spacecraft were located on the night side, around 6RE away from the THEMIS spacecraft, at around 0.5 MLT and 21MLT respectively. GOES-13 spacecraft was located slightly to the dawn-side, around 2h MLT from the THEMIS spacecraft, and

may have not been able to observe injections related to the fast flow burst. However, GOES-15 spacecraft was within 1h MLT from the 3 THEMIS spacecraft and ideally located to observe injections associated with the fast flow burst, but did not observe anything. This is an indication that pseudobreakups may only cause localized particle injections that do not penetrate very deep into the magnetosphere and hence are not observed at geosynchronous orbit. The same can be shown for the other two pseudobreakup cases (Case 3 and 5) presented in this study.

Figures 8 shows the magnetosphere response to substorm fast flow bursts, Case 2, at around 08:17 UT on 25[th] December 2015. Clearly, both GOES-13 and GOES-15 observed significant flux increases in multiple energy channels. In this case GOES-15 spacecraft was within the same MLT hour ($\sim$23MLT). GOES-15 began observing a flux increase at 08:17 across all five energy channels (40 keV, 75 keV, 150 keV, 275 keV, and 475 keV), with the most significant increases in flux observed at




low energies (e.g, 40 keV, 75 keV, and 150 keV) that was more pronounced (by up to 2 orders of magnitude). GOES-13 also

observed an increase in flux across all energy channels a few minutes later, despite being located more than 3h MLT away on the dawn-side. This shows that substorm fast flow bursts are more likely to produce a strong inner magnetosphere response. The other two substorm fast flow bursts also show strong magnetosphere responses that were similarly observed by the GOES spacecraft. It is worth noting that we studied flux observations for other geosynchronous satellites that were ideally located on the night side to observe such injections at the time of these fast flow bursts, such as the Los Alamos National Laboratory

(LANL) satellites, for which similar particle injections were observed.

In addition, to understand the difference in the magnetospheric response between substorm fast flow bursts and pseudo-breakups, we studied the curvature force density (F) Li et al. (2011); Palin et al. (2012), estimated based on equatorial pressure gradient (panels h of Figures 7 and 8), and the magnetic lobe ($B_{lobe}$) (panels g of Figures 7 and 8) using the unique configuration of the 3 THEMIS spacecraft, in close proximity, coplanar, with a normal directed along Ygsm Artemyev et al. (2019). The results show a clear and consistent difference between pseudobreakups and substorm fast flow bursts. For pseudobreakups (Figure 7), the $B_{lobe}$ decreased from ~42nT before the fast flow burst to ~34nT after fast flow burst, while there were fluctuations during the fast flow burst. The curvature force density also fluctuated, but was largely similar and relatively small before and after the fast flow burst (~50

In contrast, for substorm fast flow burst shown in Figure 8, the decrease in $B_{lobe}$ value was much more apparent as $B_{lobe}$ decreased from (~54nT) to (~34nT). Also the fluctuations consisted of larger amplitudes and higher frequencies. The curvature force density increased gradually to ~3400

## 5   Conclusion

In this study, multipoint analysis of conjugate magnetospheric and ionospheric observations were used to investigate the magnetospheric and ionospheric responses to substorm fast flow bursts and pseudobreakup events. The 3 inner THEMIS spacecraft (THEMIS A, D, and E) in situ measurements of THEMIS observations were used to select 3 pairs of fast flow bursts asso-

ciated with substorm and pseudobreakup events. These fast flow bursts were observed during close separations on the night side, beyond 6RE from the Earth in radial distance, and a magnetic local time (MLT) region of ±5 hours from local midnight. We studied in detail the magnetospheric and ionospheric response to each substorm fast flow burst and pseudobreakup event to understand what properties control the differences in the magnetosphere-ionosphere responses between substorm and pseudobreakup conditions, and how such differences lead to the different ionospheric responses.

The results show that ionospheric currents respond to both substorm fast flow bursts and pseudobreakup events, indicating that the ionosphere currents are created by plasma flow vortices in the magnetosphere for fast flow bursts that are associated with substorms and pseudobreakup events. The magnetic flux in the tail is much stronger for strong substorms and much weaker for pseudobreakup events. The $B_{lobe}$ decreases significantly (by up to 40%) for substorm fast flow bursts, but much smaller decrease in $B_{lobe}$ for pseudobreakup events. The curvature force density for pseudobreakups are much smaller than

substorm fast flow events, indicating that the pseudobreakups may not be able to penetrate deep into the inner magnetosphere.





In addition, the magnetospheric and ionospheric response to substorm fast flow bursts is stronger compared to pseudo-breakups. The pseudobreakups may only cause localized particle injections that do not penetrate very deep into the inner magnetosphere. However, more magnetic flux and energy are released during substorm fast flow bursts and hence the substorm fast flow bursts are capable of penetrating deep into the inner magnetosphere and produce a much stronger magnetospheric

response. This association can help us study the properties and activity of the magnetospheric earthward flow vortices from ground data.

The authors would like to thank Anton Artemyev for useful discussions. HA, JB, JL, JW, and XC would like to acknowledge the NASA HSR grant 80NSSC18K1227. The THEMIS data are publicly available from http://themis.ssl.berkeley.edu. The SECS-EIC data are publicly available from http://www.igpp.ucla.edu/public/jweygand/SECS/. The SuperMAG data are publicly available from http://supermag.jhuapl.edu/. The GOES 13 and 15 electron flux data and magnetic field data are publicly available from NOAA (https://www.ngdc.noaa.gov/ stp/satellite/goes/). The solar wind and geomagnetic index data were from OMNIweb (https://omniweb.gsfc.nasa.gov/ ow$_m$$in.html$).



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

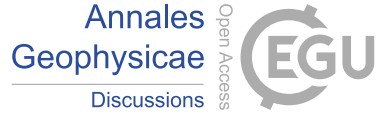

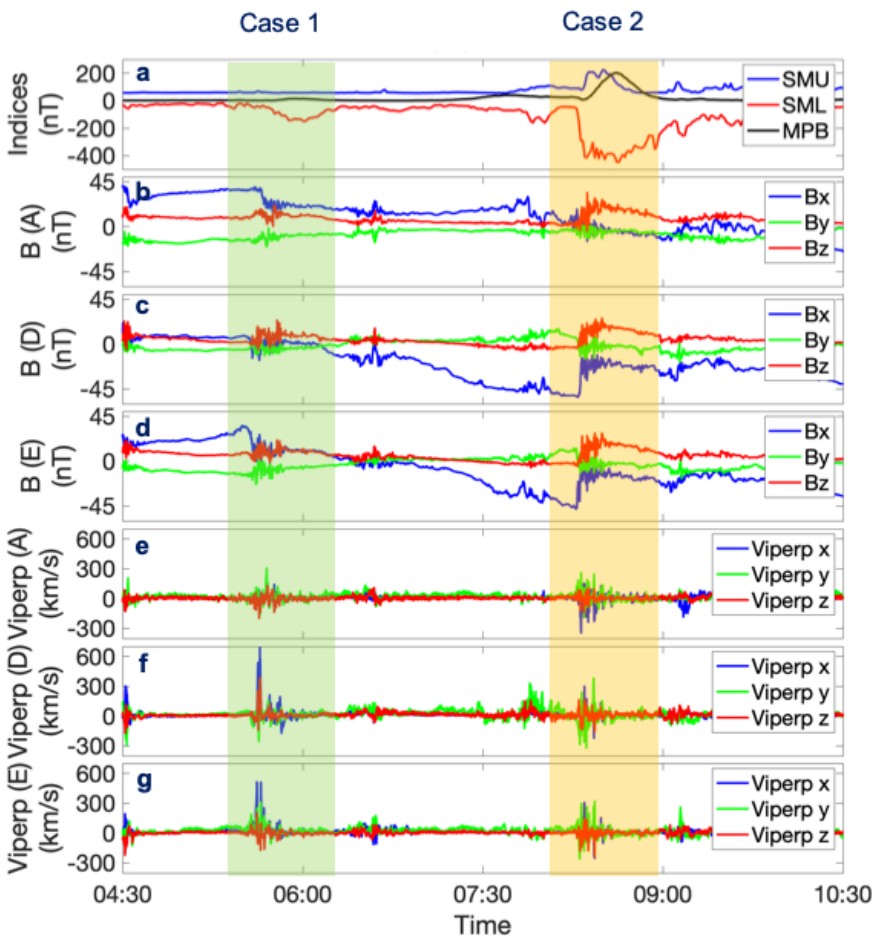

**Figure 1.** THEMIS spacecraft in situ measurements for fast flow burst Cases 1 and 2 that were observed on the 25[th] December 2015. The pseudobreakup events are highlighted in green (Case 1) while the substorm fast flow burst events are highlighted in yellow (Case 2). (a) The Auroral Electrojet Indices SMU, SML, and MPB index. (b-d) The 3-D magnetic field measured by THEMIS A, D and E respectively. (e-g) The perpendicular ion velocity measured by THEMIS A, D and E respectively.



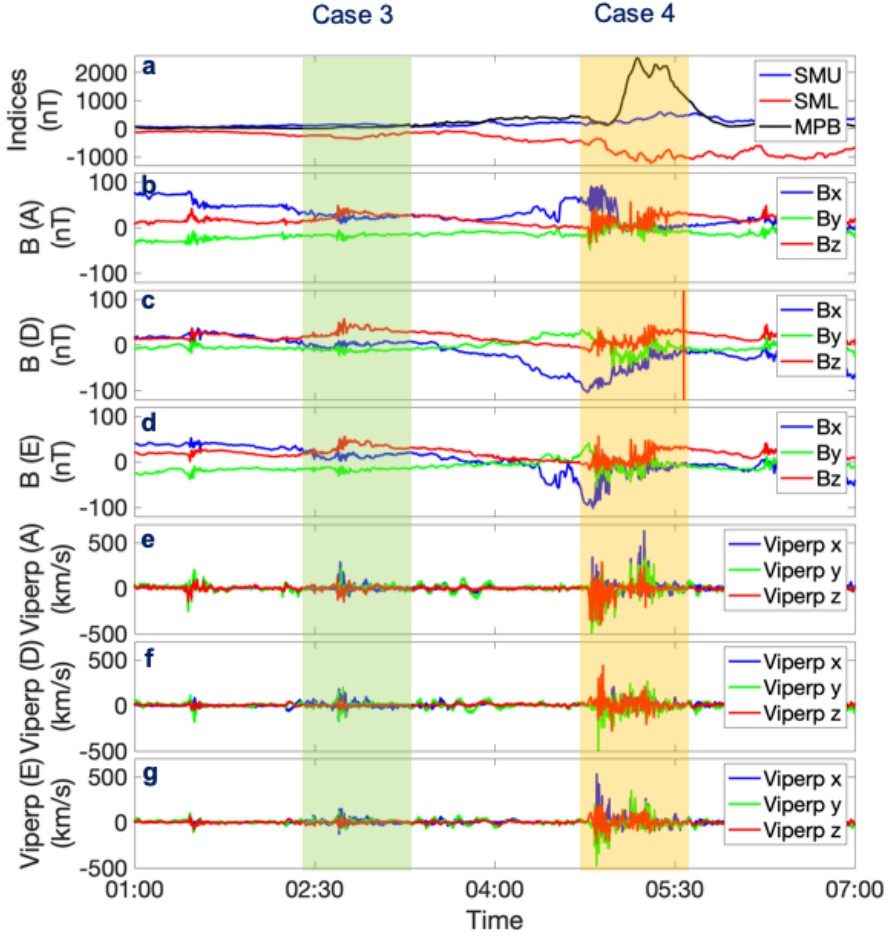

**Figure 2.** THEMIS spacecraft in situ measurements for fast flow burst Cases 3 and 4 that were observed on the 20[th] December 2015. Caption of Figure 1 applies.





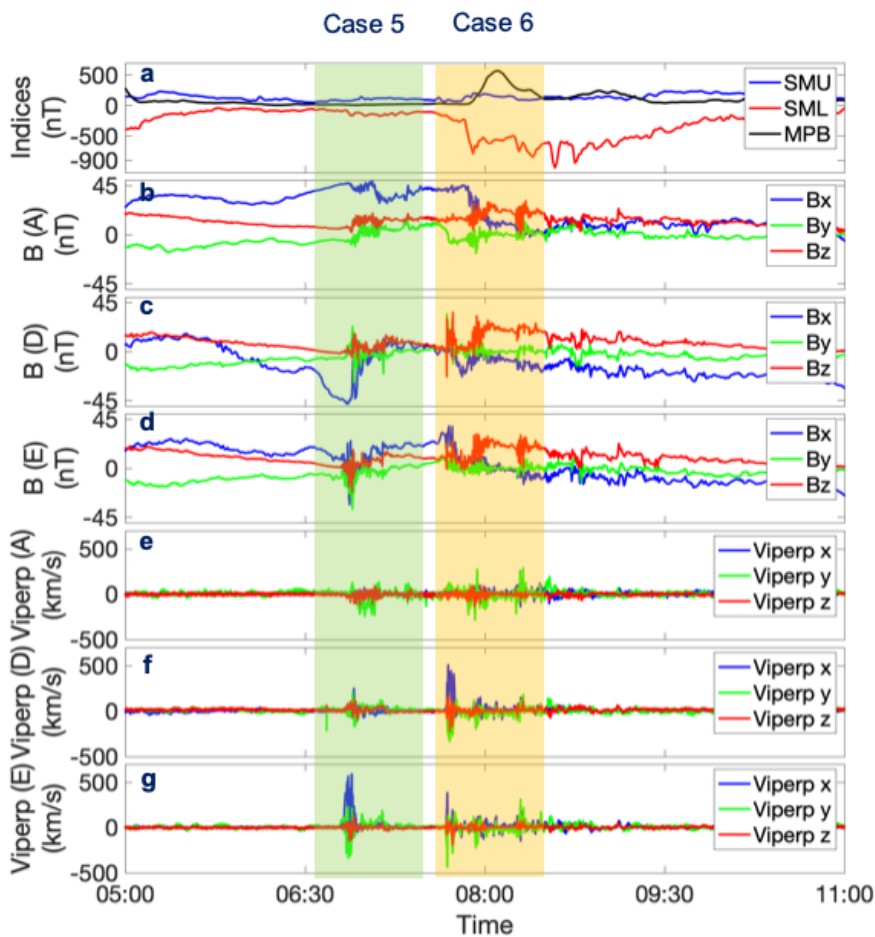

**Figure 3.** THEMIS spacecraft in situ measurements for fast flow burst Cases 5 and 6 that were observed on the 10th December 2015. Caption of Figure 1 applies.



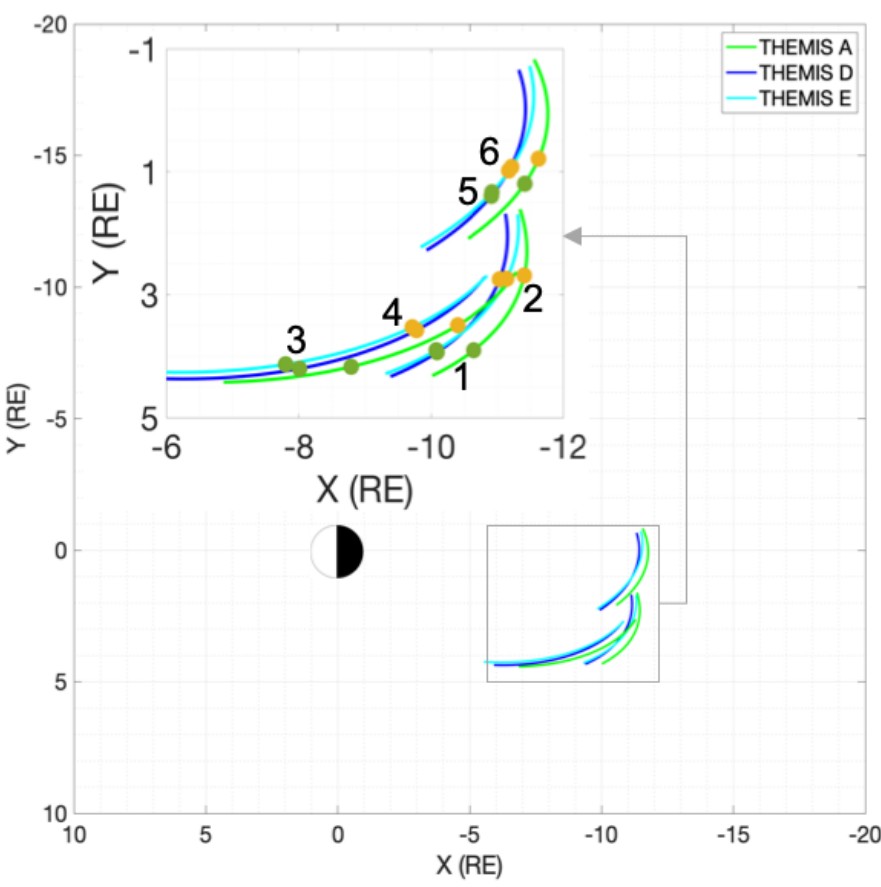

**Figure 4.** The location of the 3 THEMIS satellites (THEMIS A, D and E) GSM coordinate system for the selected fast flow burst Cases 1-6. The green and yellow dots indicate the exact location where each fast flow burst was observed along the spacecraft orbit.



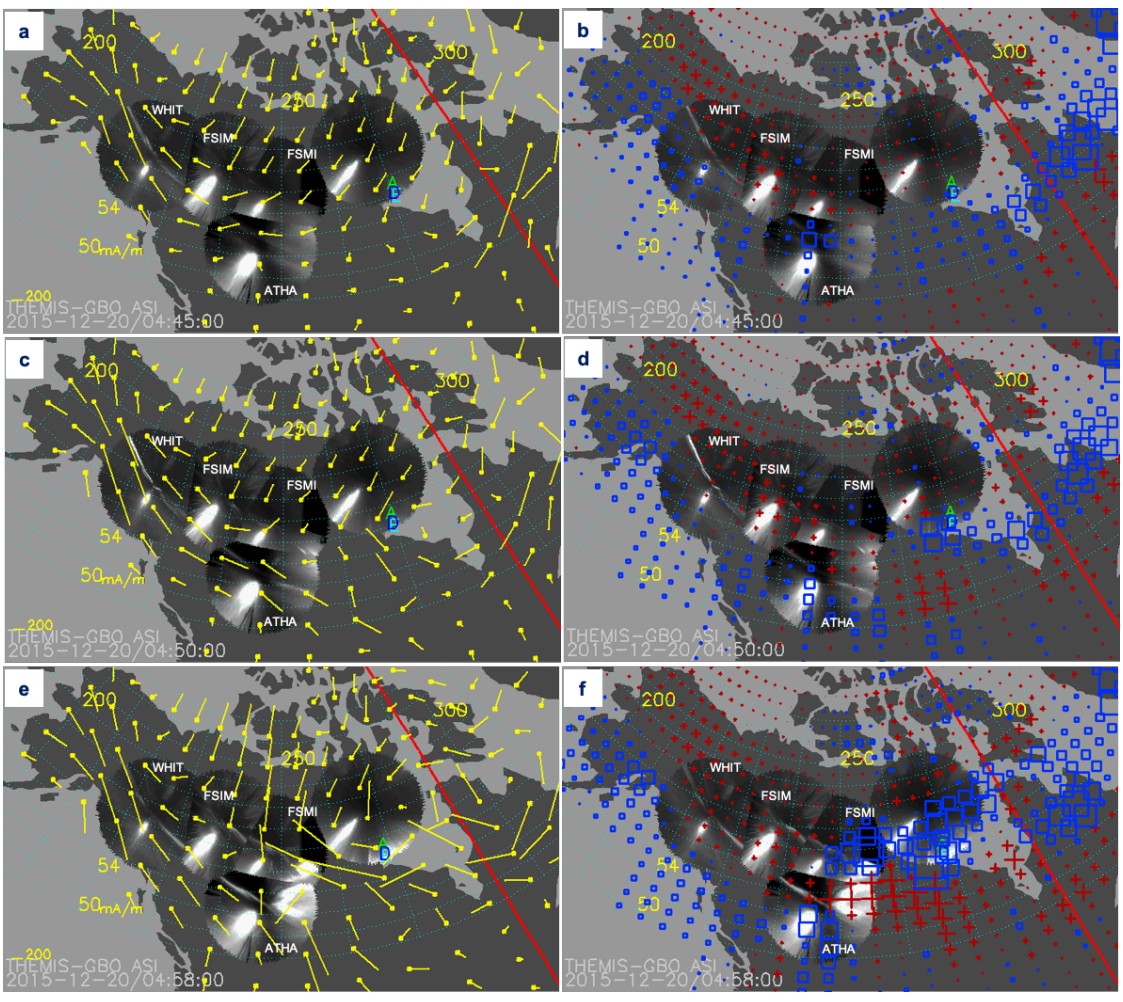

**Figure 5.** The derived equivalent ionospheric currents and current amplitudes plotted over a sequence of ASI mosaics for fast flow burst Case 4 on 20th December 2015. The yellow arrows represent the direction and the strength of the horizontal currents. The blue squares and the red plus signs show the current amplitudes. The red lines mark the midnight local time. The foot prints of the 3 THEMIS satellites at an altitude of 110km are marked by letters A (THEMIS A), D (THEMIS D), and E (THEMIS E). Each ASI field of view is approximately 800km when mapped to a 110km altitude. The snapshots show measurements (a and b) before, (c and d) during, and (e and f) after the fast flow burst.



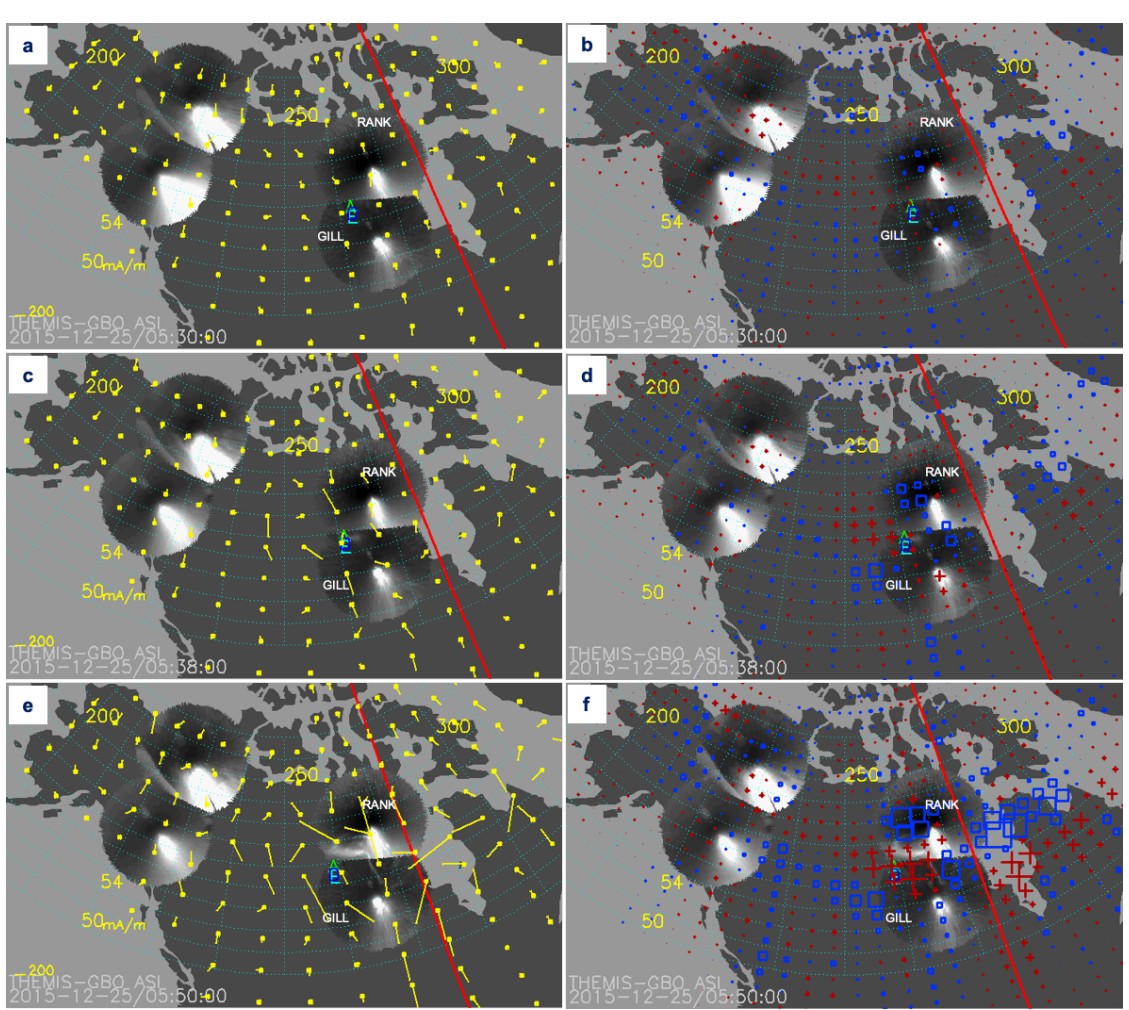

**Figure 6.** The derived equivalent ionospheric currents and current amplitudes plotted over a sequence of ASI mosaics for fast flow burst Case 1 on 25[th] December 2015. Caption of Figure 2 applies.







**Figure 7.** THEMIS spacecraft in situ measurements for pseudobreakup Case 1 on 25th December 2015. (a-c) The magnetic field measured by THEMIS A, D and E respectively. (d-f) The perpendicular ion velocity measured by THEMIS A, D and E respectively. (g) The magnetic lobe ($B_{lobe}$). (h) The curvature force density (F). (i and j) The magnetic field from GOES-13 and GOES-15 respectively. (k) Electron flux measured by GOES-13 and (l) Electron flux measured by GOES-15. The grey vertical lines mark the location of peak perpendicular ion velocity observed by each satellite associated with the fast flow bursts.







**Figure 8.** THEMIS and GOES in situ measurements for substorm fast flow burst Case 2 on 25[th] December 2015. Caption of Figure 7 applies.