# Peer review of "Multiple conjugate observations of magnetospheric fast flow bursts using THEMIS observations"

_Annales Geophysicae, 2022_

## Author Comment (AC1)

Reviewer 1

We would like to thank the reviewer for their time in reviewing this manuscript and for the valuable comments and suggestions. We have tried to implement all suggested corrections that are shown in blue text both here and in the manuscript.

First of all, we would like to apologize for the LaTex formatting errors that has led to the elimination of some text and large number of errors. We have gone through the manuscript to make sure all the errors are corrected.

This paper shows conjugate ionosphere-magnetosphere observations that suggest that substorm fast flows travel more earthward in comparison to fast flows related to pseudobreakups. Despite being a more localized event than substorms, pseduobreakup related fast flows also produce an ionospheric response but they are weaker than those produced by substorm-related fast flows.

Though the conclusions arrived in this work are not new, it strengthens them by presenting multiple conjugate ionosphere and magnetosphere measurements of fast flows and their effects. Furthermore, pairs of pseudobreakup and substorm fast-flows were selected such that they were within 5 hours of each other, attempting to make the background conditions as similar as possible.

Major

1.  Line 70-72: The paper suggests that it looks into what properties control the differences in the magnetosphere-ionosphere responses between substorm and pseudobreakup conditions, and how such differences lead to the different ionospheric responses. This goal is not completely met by the rest of the paper. Perhaps a deeper analysis of the observations pointed out in the observations section can do this goal justice.

    The main properties that we have discussed here that controls the ionospheric response to different substorm fast flows are the time varying parameters such as the current density, lobe magnetic field, curvature force density, and plasma pressure. Some of these properties, specially, current density and curvature force density were only possible to calculate thanks to the unique tail science phase configuration of the 3 THEMIS spacecraft. The results show that the magnetosphere and ionosphere response to substorm fast flow bursts are much stronger and more structured compared to pseudobreakups, which is more likely to be localized, transient, and weak in the magnetosphere. The magnetic flux in the tail is much stronger for strong substorms and much weaker for pseudobreakup events. The Blobe decreases significantly for substorm fast flow bursts compared to pseudobreakup events. The curvature force density for pseudobreakups are much smaller than substorm fast flow events, indicating that the pseudobreakups may not be able to penetrate deep into the inner magnetosphere.

2. There seems to be missing text after Lines 118 and Lines 230. Perhaps a Latex formatting error. (The line numbers are also not coherent in the pdf, so I am referring to the line numbers mentioned in the margins.)

   We do apologize for the formatting error that has led to the elimination of some text. We have now corrected those errors. We have gone through the manuscript to make sure all the errors are corrected. Thank you.

3. A claim is made at the end of the abstract and end of the conclusions: 'This association can help us study the properties and activity of the magnetospheric earthward flow vortices from ground data.' I think it'll be very useful if the authors can briefly explain how this may help future studies so that readers may immediately recognize the potential of this work.

   We thank the reviewer for pointing this out. As we know Satellite data is not always available to observe these events in the magnetosphere, whereas ground data can be readily available. Therefore, if we understand how these ionospheric currents respond to substorm fast flow bursts and pseudobreakup events, then we can determine magnetospheric conditions based on ground observations.

4. Figures: It will be very useful for the readers if the authors can label aspects of the figure with arrows and texts that are being referred to in the main text of the manuscript. This is especially needed in figures 5 and 6 to point out vortex directions and Figures 7 and 8.

   We thank the reviewer for pointing this out. We have now marked the location on the figure to make it easier for the readers. Thank you.

5. A supplementary file containing the figures that show the ionospheric response, and additional GOES measurements, for the cases not shown in the main manuscript - will go a long way to benefit the ideals of data availability and transparency.

   We agree with the reviewer and we have included all the figures in the appendix for the cases not discussed in the manuscript. We have also included the derived equivalent ionospheric currents and current amplitudes for fast flow burst cases not shown in the manuscript. Thank you.

Minor

1. Regarding the title: As the paper does not focus nor go into detailed analysis about the response of the ionospheric currents to magnetospheric fast flows, perhaps a better title for this work would be more closely tied to its novelty or conclusions. For e.g., Multiple conjugate observations of different types of magnetospheric fast flow bursts.

   We have changed the title to "Multiple conjugate observations of magnetospheric fast flow bursts using THEMIS observations".

2. In the abstract, since a major feature of this study is the 'conjugate magnetospheric and ionospheric observations', it might be useful to mention that the primary ionospheric observations were made by all-sky cameras and magnetometer-based equivalent ionospheric currents.

   We have now pointed this out in the abstract. We thank the reviewer for pointing this out.

3. Line 37: The acronym MPB - mid-latitude-positive bay should be defined here, as it's the first occurrence.

   We have now added the definition. Thank you.

4. Line 81-82: Authors say that they have analyzed 11 years of data. However, in 110, they note that the unique configuration lasted only for 3 months. Perhaps, the phrase "11 years of observations" can be omitted as it does not really reflect the final range of data used in this study.

   We agree with the reviewer. Even though we have looked into 11 years of THEMIS data, this study is primarily based on the unique configuration of the THEMIS satellites that lasted for 3 months. We have now omitted "11 years of observation". Thank you.

5. Line 152-153: The authors say that the y-coordinates of the satellites were almost the same, so all the differences in the measurements are due to separation in the (x,z) plane. I think the authors are saying that the distance between the spacecrafts in this plane does not exceed 1000 km. If so, perhaps it can be made clearer by also including an additional plot in Figure 4 of the X-Z plane as well.

   We have now updated the figure to show the location of the satellites in all 3 planes. Thank you.

   We would like to thank the reviewer for all the above comments.

---

## Author Comment (AC2)

Reviewer 2

We would like to thank the reviewer for their time in reviewing this manuscript and for the valuable comments and suggestions. We have tried to implement all suggested corrections that are shown in blue text both here and in the manuscript.

First of all, we would like to apologize for the LaTex formatting errors that has led to the elimination of some text and large number of errors. We have gone through the manuscript to make sure all the errors are corrected.

This manuscript presents 3 cases studies of pair of events, where in each pair the first event is a pseudo breakup and the second a substorm onset. The events are studied by conjugate observations in the magnetosphere and ionosphere, offered by the THEMIS satellites and the THEMIS network of magnetometers and all-sky cameras in the American sector.

The main conclusion is that the effects of substorm-associated fast flow bursts in the magnetosphere and ionosphere are much stronger and more structured compared to those that are observed during pseudo breakups. In the ionosphere intensified currents and current vortices were observed both during pseudo breakups and substorms, but they were stronger in the latter case. The magnetospheric differences between the two groups were clearly seen in the electron fluxes and changes of the lobe magnetic field.

I need to point out that the manuscript seems to have been hastily submitted, and would have benefited from a final round of polishing and checking. Now the incomplete sentences, unfinished citations and other small errors give an unnecessarily negative impression of the whole manuscript.

In summary, the manuscript presents rather interesting multipoint studies of substorms and pseudo breakups, and may be accepted for publication after some corrections and clarifications.

MAJOR COMMENTS

As noted, there are several annoying errors in the text that tell of poor quality control and lack of polishing. For example

We do apologize for the formatting error that has led to the elimination of some text. We have now corrected those errors. We have gone through the manuscript to make sure all the errors are corrected. Thank you.

- incomplete sentences missing some or several words, at least on lines 112, 119, 238,241

They are all corrected. Thank you

- use of parenthesis in the citations

We have corrected them. Thank you.

- missing citations in line 35, 66, 109

They are all fixed. Thank you

- Case 4 is not in figure 1, lines 163-165

Case 4 is shown in Figure 2. We have now corrected this. Thank you.

Taken individually the errors are reasonably minor, but their large number gives an unprofessional impression of the whole work. I recommend that you go through the manuscript very carefully before resubmitting.

Line 54: It's better to say "mostly Pedersen" and "mostly Hall", as also Hall current may have divergence and therefore connect to FACS, and Pedersen current may have some contribution to the electrojets.

We agree with the reviewer and we have made the necessary change. Thank you.

Lines 87-92. It's true that there are both curl-free and divergence-free SECS, but only the divergence-free type is used in the ground magnetometer analysis. You should clarify this point and also more carefully describe the meaning of the current amplitudes (i.e. the magnitudes of the divergence-free SECS) that are shown on the right side panels of Figs 5 and 5. In lines 190-192 and 197 you seem to identify the amplitudes with FAC, so it is necessary to list the assumptions that are involved there.

The current amplitudes are simply the current perpendicular to the ionosphere at an altitude of 100 km. Technically they are not magnitudes because they have a direction (up or down). They are not the field align currents because they are perpendicular to the ionosphere and not directly aligned with the magnetic field. This is an important point. We frequency refer to them as a proxy for the field aligned currents because it pacifies most people. Although, at the auroral regions current amplitudes are pretty close to FAC. One great reference discussing the derivation of the current amplitudes is given below. We have added this to the manuscript to clarify with some additional text.

Amm, O.,Engebretson,M.J.,Hughes,T.,Newitt,L.,Viljanen,A.,Watermann,J.,2002. A traveling convection vortex event study: Instantaneous ionospheric equivalent currents, estimation of field-aligned currents, and the role of induced currents. J.Geophys.Res.107,1334. http://dx.doi.org/10.1029/2002JA009472.

The selection criteria in Section 3 should be discussed more carefully. For example, what were the criteria for the SML index? Were the all-sky camera images used in the selection, i.e. do you require visible auroral activity in all pseudo breakups?

The substorm fast flow bursts and pseudobreakup events were selected based on the MPB index. The MPB substorm was defined as the MPB index larger than $25nT^2$. The SuperMAG Auroral Electrojet Indices (SMU and SML) were checked and plotted for convenience to show the difference for substorm fast flow bursts and pseudobreakup events. The MPB index was used mainly because it is insensitive to the localized fine structure of the electrojet and

can well capture the global substorm current wedge. We have rewritten the selection criteria for clarity. Thank you.

When discussing Figures 5-6 it would be good to mark the areas of interest to the panels, as now it is bit hard to follow which features are discussed, and should one look at the arrows on the left panels or the amplitudes on the right panels.

We thank the reviewer for pointing this out. We have now marked the location on the figures to make it easier for the reader. Thank you.

You study 3 event pairs, but detailed data are shown only for couple selected events. I recommend that you would collect the key parameters (e.g. magnitude of ionospheric currents, changes in lobe magnetic field, particle fluxes etc) from all events to a table. This would strengthen your conclusions and give the readers a firm understanding of the common features.

We agree with the reviewer, however, it was suggested by the other reviewer to include these figures in the appendix for the cases not discussed in the manuscript. We therefore ask the reviewer to advise us if a table is still recommended? Thank you.

It's bit unclear to me which results are new and which agree or disagree with previous studies. Also the implications on and future potential to "study the properties and activity of the magnetospheric earthward flow vortices" remains rather vague. I recommend that you add some discussion of these points to Section 5.

We study in detail what properties control the differences in the magnetosphere-ionosphere responses between substorm fast flow bursts and pseudobreakup events, and how such differences lead to the different ionospheric responses. The results show that the magnetosphere and ionosphere response to substorm fast flow bursts are much stronger and more structured compared to pseudobreakups, which is more likely to be localized, transient, and weak in the magnetosphere. The magnetic flux in the tail is much stronger for strong substorms and much weaker for pseudobreakup events. The Blobe decreases significantly for substorm fast flow bursts compared to pseudobreakup events. The curvature force density for pseudobreakups are much smaller than substorm fast flow events, indicating that the pseudobreakups may not be able to penetrate deep into the inner magnetosphere. The unique tail science phase configuration of the 3 THEMIS spacecraft provided us with the opportunity to determine the time varying parameters such as the current density, lobe magnetic field, curvature force density, and plasma pressure. Some of these properties, specially, current density and curvature force density were only possible to calculate thanks to this unique tail science phase configuration.

Acknowledgments:: Check the omniweb address. SuperMAG web page gives specific sentences that should be used when utilizing the SuperMAG substorm lists and the SuperMAG indexes.

Corrected. Thank you

We would like to thank the reviewer for all the above comments.

---

## Referee Report (RR1)

**Title**: The response of ionospheric currents to different types of magnetospheric fast flow bursts using THEMIS observations

**Authors**: Homayon Aryan, Jacob Bortnik, Jinxing Li, James Michael Weygand, Xiangning Chu, and Vassilis Angelopoulos

**Manuscript ID**: https://doi.org/10.5194/angeo-2022-3

The authors have answered almost all my comments in a satisfactory manner and revised the manuscript accordingly. As the other caes are included in an appendix, there is no need for the table I suggested previously. My only remaining comment, detailed below, is a technical clarification and does not affect the results or conclusions. I have no further comments, and the manuscript can be published with that minor clarification without an additional review round.

   **Technical comment**

I find the description of the relation between the SECS amplitudes and the answer to my previous comment somewhat lacking, or perhaps even misleading to the casual reader. In the SECS analysis of equivalent current from ground magnetic field, like done by Amm+Viljanen (1999) or Weygand et al. (2011), only divergence-free SECS are used. So all the current is in the ionospheric plane, there are no currents perpendicular to ionosphere (field-aligned or vertical). The amplitudes of the divergence-free SECS represent the curl of the horizontal equivalent current, and are not directly associated with any vertical current. In order to get the vertical current from the equivalent current, some assumptions must be made, like explained e.g. in Amm et al (2002):

1) Ionospheric electric field must be a potential field. This is almost always OK.

2) Field-lines must be approximately vertical. This is OK in the auroral regions.

3) Gradients of the Hall and Pedersen conductances must by aligned with the electric field. This may be fine in a statistical sense, or in some very symmetric case like the convection vorted discussed by Amm et al.

4) The ratio of Hall/Pedersen conductance must be spatially constant. This may or may not be the case. If the above assumptions are valid, then FAC = a * (curl of equivalent current), where a is the conductance ratio.

It is of course fair to say that the SECS amplitudes a proxy for the FAC, but in addition to giving the reference to Amm et al. (2002), I suggest that some more details of the required assumptions are given to the readers, who may not be familiar with the topic.

---

## Author Response (AR2)

Review

Once again we would like to thank the reviewers for their time in reviewing this manuscript and for the valuable comments and suggestions. We have no doubts that thier constructive comments has improved the quality of this manuscript. We have tried to implement all suggested corrections that are shown in blue text.

There are a couple minor edits that caught my eye, which I am noting here so that the authors may choose to modify it.

Page 4, Lines 102-103 and 120-121 are repetitive. Perhaps one of them can be removed. --> "The MPB index is used because it is insensitive to the localized fine structure of the electrojet and can well capture the global substorm current wedge"

We thank the reviewer for pointing this out. We have now removed the sentence on lines 102-103.

Page 19, Figure 5 caption: "The green circles mark thee area of discussion" change to -> ".. the area..."
Corrected

We would like to thank the reviewer for all the above comments and corrections.